# A Critical Analysis of the Rights and Obligations of the Manager of a Limited Liability Company: Managerial Legislative Basis

**Tomáš Peráček** *** and **Michal Kaššaj**

Faculty of Management, Comenius University Bratislava, Odbojarov, 10, 820 05 Bratislava, Slovakia; kassaj2@uniba.sk
*   Correspondence: tomas.peracek@fm.uniba.sk

**Abstract:** The rights and obligations of an executive as a top manager of a limited liability company seem to be a long-settled question. However, the opposite is true. We were particularly interested in the question of the rights and obligations of the manager as a statutory body of the most widespread type of business company. A very important issue is the definition of the relationship between the limited liability company and the manager. The reason for this is the fact that it is a business–legal relationship and the protection provided to the executive in relation to the business company is based only on their mutual contractual basis. In addition to the examination of managerial knowledge and skills, we focused primarily on a critical analysis of the legal definition of the rights and obligations of an executive and their responsibility towards a limited liability company. As part of our research, we analyzed extensive jurisprudence, which completed our understanding of the concept of an executive and also defined the framework of not only their actions, but especially their rights and obligations. To achieve our goal, we used several scientific methods designed for the study of law, such as analysis, synthesis, comparison, deduction, and description. We critically evaluated the results of our research and compared the development of Slovak and Czech jurisprudence in the context of its influence on the investigated issue. At the same time, we answered the research question of whether legislative intervention is necessary for the already existing rights and obligations of a manager in relation to their limited liability company. This analysis of the rights and obligations of the manager of a limited liability company has several implications for both managers and companies as a whole, such as managerial autonomy, accountability, responsibility, and the balance of power. The research findings highlighted the significant decision making authority granted to managers. The obligations identified in the analysis emphasized the need for managers to act responsibly and be accountable for their actions. The rights and obligations of managers need to be balanced with the interests of other stakeholders, particularly the company's members. In conclusion, the critical analysis of the rights and obligations of the manager of a limited liability company, based on the managerial legislative basis, revealed that managers possess decision making authority, profit distribution rights, limited liability protection, and entitlement to compensation. However, they are also obligated to fulfill fiduciary duties, comply with laws and regulations, maintain proper records, and exercise due care. The research underscored the significance of managerial autonomy, accountability, and a balanced exercise of power in a limited liability company. By understanding and adhering to their rights and obligations, managers can effectively navigate their roles while contributing to the success and sustainability of the company

**Keywords:** commercial code; manager; limited liability company; rights and obligations

## 1. Introduction

Are the rights and obligations of managers in limited liability companies adequately regulated? A critical analysis of the managerial legislative basis the reveals strengths,

weaknesses, and potential gaps in the existing framework. A limited liability company acts through managers. The fact that limited liability companies are the most widespread business companies is understandable for various objective reasons. This form of a business company, according to Horvat et al. (2017), most closely copies the dynamic trend of the market economy and maintains a simple structure, but, at the same time, eliminates the negative impact of business risk on entrepreneurs, because an entrepreneur is liable for the company's obligations only up to when the amount of their unpaid deposit entered in the commercial register.

From the very nature of a limited liability company as a legal entity, it is understandable that it cannot act alone, but acts through certain persons. With this, according to Mitterpachova et al. (2019), legal regulation has had to effectively settle so that, on the one hand, the company can act and appear in legal relations, and, at the same time, that this company is also protected from the potential personal interests of the persons who will act on its behalf. The solution to the given problem was the creation of a statutory body—an executive who represents a limited liability company, while their actions are considered to be the actions of a limited liability company. The statutory body is limited by various provisions, in order not to abuse its position. It is at least one natural person acting in the position of executive as an obligatory body, without which, the company cannot exist.

According to Sararu (2023), their main duty is to act for the company externally and, at the same time, manage it. The above-mentioned duties are only part of the duties that a manager must perform. From the theory of law, we also know that, in contrast to every obligation, there is also a certain right, and thus the executive has a whole range of rights. Due to the fact that the issue of the rights and obligations of an executive as a top manager is a topic that not only legal theory deals with to a lesser extent, our ambition is to fill this gap by examining the rights and obligations of an executive, defining the managerial skills that every executive should have and, at the same time, explain upon what legal basis the relationship between the limited liability company and the manager can function.

The aim of the study is to conduct a critical analysis of the rights and duties of the managing director of a limited liability company based on the managerial legislative base. Specifically, this research seeks to examine the legal framework governing the managerial function in limited liability companies, enriched by the case law of judicial authorities, and to assess the extent to which the rights and duties of managers are defined and enforced by legislative provisions. The study will look at the relevant statutes, regulations, and legal precedents to provide a comprehensive understanding of the managerial legislative basis and its implications for managers in the discharge of their duties and responsibilities. By examining the legal framework, the research aims to identify potential gaps, problems, or areas for improvement that could enhance the effectiveness and accountability of managers in limited liability companies.

The research problem in this study is to critically analyze the rights and obligations of the manager of a limited liability company within the managerial legislative basis. The research objective is to evaluate the existing legislative framework governing the rights and obligations of the manager in a limited liability company and identify any strengths, weaknesses, or gaps in these regulations. The research question guiding this study is: Are the rights and obligations of the manager of a limited liability company adequately addressed and regulated within the current managerial legislative basis?

In our scientific study, in addition to the analysis of the manager as a statutory body of a limited liability company, we focus primarily on a critical analysis of the rights and obligations of this manager.

In order to achieve the main goal, we also had to determine sub-goals, namely the analysis of:

- The managerial knowledge and skills of the executive,
- The individual provisions of the Commercial Code, the Civil Code, and the Trade Act.

At the end of our study, we will critically evaluate the results of our investigation, compare the development of Slovak and Czech jurisprudence and its impact on the subject

under investigation, and answer the research question of whether the position of an executive as a statutory body of a limited liability company is sufficiently regulated or whether a significant legislative intervention in the Slovak legal system is necessary.

## 2. Literature Review

The manager acts for the limited liability company as a legal entity. According to Bondareva and Zatrochova (2014), the actions of the executive are considered to be the actions of the company itself. The actions of the manager are, therefore, a manifestation of the will of the limited liability company. According to the legal diction, the statutory body is one or more executives. However, the exact number of these executives remains at the discretion of the company's founders (Kuciuk 2021; Gazacu 2023).

As stated by Cajka and Abrham (2019), the Commercial Code stipulates that, if the founders of the company have elected several managers, each of them is entitled to act on behalf of the company independently, unless the articles of association stipulate otherwise. According to them, the last part of the sentence, after the comma, is very important. It defines the possibility to modify the actions of individual managers in the social contract. This means that, for example, in the case of two executives, the articles of association may specify that both of these executives act together. Limitations of this executive authority should be distinguished from the way of acting of executives.

As Funta (2023) further emphasizes, the number of executives also depends on the size of the company. The larger the business company, the greater the number of executives that should be involved with it. The main idea is that a larger number of executives will ensure greater flexibility for the company during the normal operation of the given business company. On the other hand, however, a larger number of executives increases the risk that they will conclude contracts that are disadvantageous for the business company. Gregusova et al. (2016) only partially agrees with this opinion. Gregusova et al. (2016) points out the fact that such contracts can be concluded by a company with only one manager, but the partners are better able to control a smaller number of managers.

The Commercial Code in §133 par. 2 defines that an executor can only be a natural person who, at the time of entry into the commercial register, is not listed as an oblige in the register of authorizations for execution under a special law. In this, Dušek (2018) finds a difference compared to the partners in a limited liability company. A partner in a limited liability company can be both a legal entity and a natural person. In the case of the performance of executive function, the law allows the appointment of only a natural person. In principle, it does not matter whether the manager will be appointed from among the partners or from persons who are outside the company. Based on practical experience, according to Kubickova et al. (2022), it is more appropriate if the manager is also a partner the of limited liability company. According to them, this is mainly due to the fact that, in this case, they also have a certain psychological relationship with society, which consists of a higher commitment to its benefit. On the other hand, according to Čajkova et al. (2023) a situation arises when a partner does not have the sufficient managerial skills and, in this case, it is more appropriate to have an executive from among persons who are outside the company. On the one hand, it is very important that the manager is interested in the prosperity of the company, but on the other hand, they must also have certain management skills.

In conclusion, the literature on the rights and obligations of the manager of a limited liability company highlights several key themes and gaps. The manager, as the legal representative of the company, acts on behalf of the company itself, and their actions are considered to be a manifestation of the company's will. The number of these managers can vary and it is typically determined by the founders of the company.

The Commercial Code allows for multiple managers to act independently, unless the articles of association specify otherwise. The ability to modify the actions of individual managers through these articles of association is an important consideration. The size of the company can influence the number of executives, with larger companies generally having

a greater number of managers to ensure flexibility in their normal operations. However, a larger number of executives also increases the risk of unfavorable contracts for the company. There is a difference between limiting executive authority and the way that managers act within the company, which should be carefully distinguished.

The Commercial Code specifies that only natural persons can serve as managers, and they must not be listed as obligees in the register of authorizations for execution under a special law. Managers in a limited liability company can be either legal entities or natural persons, but only natural persons can be appointed as managers. The literature suggests that having a manager who is also a partner of the company may be more appropriate due to their psychological commitment to the company's benefit. However, it is also acknowledged that, in some cases, an executive from outside the company may be more suitable if the partner lacks the sufficient managerial skills.

Overall, the existing literature provides insights into the rights and obligations of managers in limited liability companies, highlighting the importance of the articles of association, the number of executives, and the qualifications and commitment of the manager. However, there are still gaps in the literature, particularly in terms of detailed guidelines or best practices for selecting these managers and balancing their rights and obligations. Further research is needed to address these gaps and provide more comprehensive guidance for managers and companies operating under limited liability structures.

## 3. Materials and Methods

We want to achieve the set main goal and sub-goals mainly by thoroughly examining the relevant Slovak and Czech legislation, as well as jurisprudence. Other focal sources of knowledge were the professional and scientific literary sources contained in the Web of Science and Scopus databases. The main sources of knowledge upon which our scientific study was based are:

- Act of the National Assembly of the Czechoslovak Socialist Republic No. 40/1964 Coll. Civil Code as amended (National Assembly of the Czechoslovak Socialist Republic 1964);
- Act of the Federal Assembly of the Czech and Slovak Federative Republic No. 455/1991 Coll. on trade entrepreneurship (Trade Act) as amended (Federal Assembly of the Czech and Slovak Federal Republic 1991a);
- Act of the Federal Assembly of the Czech and Slovak Federative Republic No. 513/1991 Coll. Commercial Code as amended (Federal Assembly of the Czech and Slovak Federal Republic 1991b).

We used the methods of logic to gain knowledge about the law. The methods of logic are universal and can be used in all sciences. These methods of logic determine the rules of human judgment based on many years of experience. Their observance ensures the correctness of thinking, truthfulness of thinking, and its orderliness. When designing this scientific study, we also applied the method of abstraction, which ensures the unification of two thought processes and then separates the most important from them. In the article, we used the method of abstraction for legal provisions that are directly related to each other, combining them into one logical whole and then abstracting the most important from them. The synthesis method found its application in the division of the text into individual parts, followed by the analysis of these parts, and, finally, their connection into a single whole. The methods of synthesis and analysis were used practically throughout the article, for example, for the division of individual rights and obligations into smaller units, whereby these smaller units were subsequently analyzed, finally combined into one larger unit, and conclusions were drawn from it.

Another category of methods of knowledge are scientific methods of knowledge. The use of the analytical method is essential when analyzing the current legal status of the position and powers of the manager in a limited liability company. The aim of using this method was to obtain several opinions on the given issue of legislation and interpretations of individual concepts. Additionally, in the thesis, the method of comparison was used when comparing the different legal regulations in different legal branches. This method

allowed us to obtain the similarities or differences in these individual legal regulations in various legal branches. As part of learning about the current legal status de lege lata, we also used the Judgments of both Slovak and Czech courts, as well as doctrinal interpretation.

In conducting our analysis and comparison, we employed several methods and criteria to ensure a comprehensive and rigorous examination of the rights and obligations of the manager in a limited liability company. The selection criteria for legal provisions were based on their relevance to the topic and their applicability within the Slovak legal system. The primary legal sources considered in our study were the Civil Code and the Commercial Code.

To gather knowledge and insights, we extensively reviewed the relevant Slovak and Czech legislation, as well as jurisprudence. We relied on professional and scientific literary sources. These sources provided us with valuable insights and perspectives on the subject matter.

The individual legal provisions were selected taking into account the possible problems in their application practice. The provisions were selected on the basis of a critical evaluation of the questionable provisions, also in the light of recent case law on these matters. In general, the more judgments referring to a given provision, the more debatable and ambiguous the provision is; therefore, it was appropriate to select these provisions for our critical analysis. Consequently, we selected only the most important judgments that have not yet been overturned by a higher court. Once we came to the judgments that have been issued by the courts at the same instance, we incorporated these into our analysis and tried to draw conclusions from them.

## 4. Results

### 4.1. Managerial Skills of the Executive

Every manager must have managerial skills if they want their company to be successful. Executives are also top managers, they are responsible for strategic decisions, they act on behalf of the company, they approve the actions of their subordinates and, last but not least, they are also responsible for the actions of their subordinates and, ultimately, they also have responsibility for whether the company is successful.

The manager is a high position within a company, which is, financially, very well evaluated. According to the data of the Statistical Office of the Slovak Republic, which monitors the development of salaries in the Slovak Republic, the salary for the position of manager (we can also talk about the position of general director, although it is not the same, but the salary conditions are approximately the same) ranges from EUR 2200 to EUR 11,700. This salary range also indicates to us that the responsibility in this position is high. We only remind you that the average monthly gross salary in the Slovak Republic for the third quarter of 2020 was EUR 1113.

So, what managerial skills must the manager master in order for the company to prosper? According to several theorists (Chochia and Kerikmae 2018), there are four basic abilities required. At the same time, the authors state that these are management functions. They are planning, organizing, leading people, and controlling. As for planning, the manager decides which activities will be carried out and when these activities will be carried out. Some activities, according to Jankelova et al. (2021), need to be carried out in the near future, and some can withstand postponement. It is very important that the manager knows when which activity needs to be performed, because sanctions may result from a failure to fulfill the given obligation. They cite filing tax returns as an example. If the manager does not plan the work on these tax returns well, as a result, the company will not fulfill its obligations to the state administration body and a sanction will be applied. The second ability that a good manager (executive) must master is organizing. According to Majercakova and Mittelman (2016), the term organizing should be understood as the creation of such a structure of relations between individual employees, on the basis of which, it will be feasible to fulfill the goals that the manager has already planned in advance. In practice, this means that, in the case of this ability, the executive as a top

manager must create an organizational structure and, at the same time, coordinate the activities of individual employees. Another skill is leading people. In the context of this ability, the executive directs and motivates employees to such an extent that their work leads to the achievement of the desired goals. The fourth basic skill that every good manager must master is control. As part of control, the manager monitors the course of the individual activities performed in the company and compares them with the desired state. In the case of the proper functioning of this process, the manager should not interfere in these activities. However, if certain discrepancies are discovered, it is necessary to correct them towards the set goals. In principle, the manager should therefore control all the activities and processes that take place within the organization. However, if we are talking about large commercial companies, then, according to Horvathova and Cajkova (2019), detailed control is not possible in this case and it is therefore necessary to delegate part of your powers to line managers. However, we cannot consider this calculation of managerial skills to be exhaustive. These are only the basic qualities that, according to the knowledge of several theorists, a successful manager should have (Matic and Mirica 2022).

*4.2. General Conditions for the Performance of the Executive Function*

In addition to the qualities and abilities that a manager should have, but does not have to have, the legislation sets the conditions that they must adhere to. Without these conditions, a natural person cannot perform the function of executive. We have already stated the conditions that a manager should have (but does not have to). It is therefore a legal–economic relationship to the business company, as well as a psychological relationship to the business company, with various characteristics (organization, the management of people, planning, and control).

The conditions that a manager must meet are not exhaustively listed in the Commercial Code. They are also found in the Trade Act, specifically in §6. From the title of the law, it may appear to us that this law applies only to tradesmen, yet the executive is not a tradesman. However, §6 paragraph 3 of the Trade Act stipulates that the general conditions for a Slovak legal entity must be met by a natural person, or persons who are its statutory body. This means that the Trade Act defines the requirements placed on the person of the manager in this form. Without meeting these conditions, the manager cannot be appointed to the position. The general conditions for the performance of executive function are:

1. Attainment of the age of 18 years—this condition is absolute, that is, no person under 18 years of age can, under any circumstances, be an executive. This absolute condition of 18 years of age should not be confused with the concept of majority. Although according to §8 of the Civil Code, majority is attained upon reaching the age of 18, at the same time, majority can also be reached before the eighteenth year of life by entering into marriage. This means that it is possible for an adult to be only 17 years old. This is exactly what the legislators wanted to prevent in relation to the person of the executive and therefore set the absolute condition of 18 years of age, and not the condition of majority.

2. Capacity for legal acts—in this case, the legislator requires that the executive has the full capacity for legal acts. Today, it is no longer possible to deprive a person of legal capacity; in this context, §10 paragraph 1 of the Civil Code is an obsolete legal norm. However, it is still possible to limit the capacity for these legal acts. The condition of full capacity for legal acts points to exactly this situation. In principle, therefore, the executive cannot be even partially limited in their capacity for legal acts. If they are limited in their capacity for legal acts, they cannot become an executive. This is also an absolute condition.

3. Integrity—This condition is considered relative, in theory. This is mainly due to the fact that a person who has either not committed any crime or has committed a crime other than that specified in §6 paragraph 2 of the Trade Act is considered to be blameless. The criminal acts mentioned in the given paragraph are economic crimes, crimes against property, or other crimes committed intentionally, the essence of which is related to the object of business, if it is viewed as if it has not been convicted (Funta and Ondria 2021).

4. The fourth condition is the prohibition of exclusion from the executive function. §13a of the Commercial Code establishes which person may not perform the function of a statutory body in relation to a decision against that person. This is about the creation of a so-called "disqualification register". The goal of this register is to eliminate the access of persons who do not fulfill their obligations in accordance with the company's interests, as a result of which, they create various harmful consequences, both for the company in question and for third parties. Likewise, the actions of these persons have a negative impact on the overall business environment and it is therefore undesirable for them to continue to hold such a high position. The decision to include a person in the register of disqualifications is issued either by a court or another body whose decision is reviewable by a court, as it is a substantial interference with the rights and interests of natural persons protected by law. In this case, however, even such interference with these rights is tolerated by society, as the public interest (a better business environment) outweighs the rights of this natural person. This public register is available online on the website of the Ministry of Justice of the Slovak Republic, so everyone can access it. There are currently 326 persons on this register. If a natural person is already listed in this register, they may not perform the function of a member of a statutory body for the period specified in the decision or for a period of three years from the date of validity of the decision. The fulfillment of this condition is verified by a lustration in the disqualification register as part of the registration of a limited liability company. This lustration is carried out by the registry court, based on the provisions of §7 paragraph 18 of the Act on the Commercial Register)

If a natural person meets the above conditions, they can be appointed to the position of executive (Janakova and Zatrochova 2019). There are two ways in which the function of executive can arise. The first way is an agreement of the partners before the first registration of the company in the commercial register. This agreement is part of the partnership agreement, as the appointment of the manager and the manner of the manager's actions are a mandatory part of this contract. The second method no longer concerns the initial registration of the company, but concerns the appointment of an executive during the operation of the company. According to Patakyova and Gramblickova (2020), during the operation of a commercial company, the adjustment of the position of the manager is resolved by a general meeting, based on §125 paragraph 1 letter f) of the Commercial Code.

*4.3. Limitation of Executive Authorization and Determination of the Manner of Action of the Executive*

It is essential to distinguish between these two concepts. In the first case, i.e., a limitation of executive authority, this is a narrowing of the scope of legal executive authority. In general, an executive has the right to perform all legal acts. In this way, we can limit this authorization only to certain actions, that is, we can narrow its scope. Any limitation of executive authority is decided either by a general meeting or by the articles of association. It should also be mentioned that these are internal/internal restrictions on executive powers and are not externally effective against third parties (Nováčková et al. 2023).

As an example, according to Matejkova and Pavelek (2020), we can cite a situation with four managers, where the social contract determines that each manager can act independently (determining the method of the manager's actions), but for the conclusion of a contract, the fulfillment of which is greater than EUR 30,000, the consent of all the executives is required.

In principle, the Commercial Code does not stipulate how a given internal restriction should sound, which means that, in this case, legislators leave it to the will of individual subjects. However, the last sentence of §133 paragraph 3 of the Commercial Code and §13 paragraph 4 of the Commercial Code (Supreme Court of the Slovak Republic 2015) are very important in relation to third parties. These paragraphs stipulate that the internal limitations of executive powers have no legal effects in relation to third parties. This is mainly based on the fact that a third party does not have the opportunity to know about the internal documents of a given business company, and the business register does not publish

these limitations of executive authority. However, §13 paragraph 4 of the Commercial Code stipulates that, even if these restrictions are published, they are not effective against third parties. In relation to a business company, this means that even the actions of the executive in violation of the internal regulations bind the given business company (Peracek 2022).

Determining the manner of action of a statutory body is a different concept. This procedure is determined by the social contract and can only be changed by decisions made in a general meeting. The manner in which the manager can act is regulated in the articles of association, but, at the same time, these data are publicly available in the Commercial Register of the Slovak Republic, because these are among the mandatory published data. With concrete examples selected from the Commercial Register of the Slovak Republic, it is possible to cite a situation where a manager has acted contrary to the manner stated in the commercial register—such a legal act is invalid. This is due to the fact that the legal act was not performed in the way that the statutory body of the company should act. This opinion is also supported by the established decision making practices of the Supreme Court of the Czech Republic (Judgment of the Supreme Court of the Czech Republic of 12 June 2001, No. 29 Cdo 695/2000; Judgment of the Supreme Court of the Czech Republic of 20 August 2002, No. 29 Odo 198/2002 Supreme Court of the Czech Republic 2001, 2002). In the context of the jurisprudence of the Supreme Court of the Czech Republic, it is still necessary to refer to Ruling No. 29 Cdo 301/2010 (Supreme Court of the Czech Republic). This jurisprudence deals with the situation where the articles of association determine the methods of action (not the limitation of executive authority) and, according to them, the method of action of the executive was entered into the commercial register so that each of the managers is entitled to act independently in business cases, the value of which does not exceed the sum of CZK 40,000; in other cases, both managers must act together. The Supreme Court leaned towards the interpretation that, concerning the joint action of managers, this is about determining the method of action on behalf of the company, according to §133 paragraph 1 of the Commercial Code, and not a limitation of executive authority, in accordance with §133 paragraph 3 of the Commercial Code. On the basis of the cited decision, it would be possible to state in the partnership agreement that the managers can act independently up to a certain amount and must act together above this amount. In principle, this would avoid the use of a limitation of executive authority. In the same way, a limited liability company would be exposed to a smaller risk, since, in the case of actions contrary to internal regulations, it must claim compensation from the manager or, if the damage did not occur, the company cannot do anything unless it has a contractual penalty agreed upon by internal agreements between the company and the executive.

However, according to the latest amendment to the Commercial Code, effective from 1 October 2020, if it is an entrepreneur who is registered in the commercial register, the limitation of the statutory body, according to the first sentence, is not registered in the commercial register. The aim of this legislation is precisely to prevent the circumvention of the institute of a limitation of executive authority and, ultimately, to create a greater sense of legal certainty. In the same way, the deficiency in the area of registry courts is eliminated, where some registry courts have refused to enter such "restriction of executive authority in the commercial register" and some have not had a problem with it. Companies that have such restrictions entered into the commercial register are obliged to modify these entries when submitting their next proposal for the entry of changes to the entered data, no later than 30 September 2021. This means that the legislator, in this case, has "broken" the jurisprudence of the Supreme Court of the Czech Republic.

### 4.4. Authority of the Executive

Executives in a limited liability company have two types of powers:

- The competence of the statutory body—acting on behalf of the company externally in all matters. This competence cannot be limited with effects towards third parties (for example, representing the company before various bodies and entering into contractual relationships on behalf of the company, etc.).

- The competence in the field of business management—in this case, the manager decides on the internal affairs of the limited liability company. This principle applies when making decisions about the company's business management, giving managers general powers. (Mamojka and Ivan 2016).

The general competence of executives defines the range of authorizations with which they can act upon within a company. Executives, on the basis of general authority, can act upon and make decisions about all matters. However, there are certain exceptions, for example when the law, the social contract, or the general assembly entrusts such decisions to another body. As long as the authority of the executives is not limited in the social contract, it is valid that they decide on issues of company development and also on the business policy of the company.

As for the scope of business management, neither the social contract nor any other document can, according to Jankelová et al. (2018), authorize only one or only one of several executives of a company. The authority in the field of business management belongs to all executives compulsorily. §134 of the Commercial Code defines the method of making decisions as business management. The decisions about the business management of the company, which fall under the competence of the executives, require the consent of the majority of the executives, if the articles of association do not specify a higher number of votes. In the case of a company that has only one manager, this section does not apply, as the manager will make the decisions themself. This applies only to companies with two or more directors. This provision also manifests itself in a certain form of the limited dispositional principle, since the social contract can adjust the number of votes, but can only adjust it upward, i.e., only a higher number of votes. If a lower number of votes was stated in the partnership agreement, it would be an illegal provision of the partnership

The legislation thus defines the way of making decisions about business management, but does not regulate the concept of business management itself. The concept of business management cannot be found in any legal regulation, but legal doctrine and jurisprudence have gradually shaped this concept. Patakyova and Gramblickova (2020) states that the term business management can include decision making about issues of an organizational nature, technical issues, issues of the internal operation of the company, the business intention of the company, and so on. In the context of the definition of the term business management, it is necessary to refer to the jurisprudence of the Supreme Court of the Czech Republic. According to the judgment of 25 May 2004, no. 29 Cdo 479/2003, the concept of business management in legal theory is predominantly understood as company management, i.e., mainly organizing and managing business activities, including deciding on business plans (Supreme Court of the Czech Republic 2004).

It is true that decisions about business management can subsequently be implemented through certain legal actions (the conclusion of a contract). A specific legal act is the consequence of a decision about business management. As an example, we can cite a situation where executives decide within the business management that the company will not enter into contracts with a certain company. In the case of a violation of this decision by any of the executives, the company will be entitled to compensation from the executive who violated this decision. This right to compensation for damage can be claimed on behalf of the company by any of the managers, but also by the partner.

*4.5. Rights and Obligations of Executives*

The manager, as one of the most important bodies in a limited liability company, has a relatively wide catalog of rights and obligations. The Commercial Code does not provide an exhaustive calculation of these rights and obligations of the manager. The rights and obligations of the manager are regulated in the general provisions in §133 to §136 of the Commercial Code on the manager. We also find that there are modifications according to the type of contract, which is governed by, e.g., mandate contracts—§566 to §576 of the Commercial Code. Last but not least, some rights and obligations can be found in the first and fifth chapters of the Commercial Code. Individual rights and obligations

are therefore literally "scattered" throughout the Commercial Code. As an example of an orderly calculation of rights, albeit from public law, we can cite §231 of the Criminal Code, where the powers of a prosecutor are defined in individual points.

Among the duties of an executive in a limited liability company are:

(a) Ensuring the proper maintenance of the prescribed records and accounting (§135(1) of the Commercial Code)

(b) Submitting to the general assembly for the approval of regular or extraordinary individual financial statements and proposals for the distribution of the profits or payments of losses (§135, paragraph 2 of the Commercial Code)

(c) Submitting an annual report if required by a special law (§135 paragraph 2 of the Commercial Code; §19 and §20 of the Accounting Act)

(d) Keeping a list of partners (§135 paragraph 1 of the Commercial Code)

(e) Informing partners about the company's affairs (§135(1) of the Commercial Code)

(f) Complying with legal conditions in the case of the acquisition of property by the company from its founder or partner (§59a of the Commercial Code)

(g) Notifying the company or its bodies of the measures that must be taken to avert imminent damage in the event of a termination of the performance of the duties of the company's executive, if the company is threatened with damage (§66, paragraph 2 of the Commercial Code)

(h) Notifying the registry court without undue delay of the repayment of the entire deposit of each partner (§113(1) of the Commercial Code)

(i) Submitting a proposal to register the company in the commercial register (§112 of the Commercial Code)

(j) Requesting the payment of late payment interest on behalf of the company from those partners who have not repaid their deposit (§113(2) of the Commercial Code)

(k) Calling on a partner who is in arrears with the payment of their deposit to fulfill their obligation within a period that must not be shorter than three months (§113, paragraph 3 of the Commercial Code)

(l) Supervising the payment of profit shares to the partners (§123 of the Commercial Code)

(m) Deciding on the use of the reserve fund (§124(2) of the Commercial Code)

(n) Convening a general meeting at least once a year (§128 paragraph 1 of the Commercial Code)

(o) Notifying shareholders about the date and program of this general meeting (§129(1) of the Commercial Code)

(p) Announcing the results of voting per rollam (§130 of the Commercial Code)

(q) Adhering to the manner of the manager's actions (§133 paragraph 1 of the Commercial Code.

(r) Complying with restrictions on executive authorizations (§133(3) of the Commercial Code)

(s) Performing their duties with professional care and in accordance with the interests of the company (§135a of the Commercial Code)

(t) Complying with the prohibition of competition (§136 of the Commercial Code)

(u) Announcing the proceedings of general meetings and granting the floor at any time when requested by the members of the supervisory board (§140 paragraph 1 of the Commercial Code)

(v) Preparing the complete wording of the partnership agreement after each change (§141 paragraph 3 of the Commercial Code)

(w) Compensating the company for any damage they have caused by breaching their duties while performing the function of executive (§135a, paragraph 2 of the Commercial Code)

(x) Satisfying creditors' claims if they cannot satisfy their claim from the company's assets; the company is entitled to compensation for damages against the executive (§135a, paragraph 5 of the Commercial Code)

(y)　Publishing reductions in the company's share capital (§147 paragraph 1 of the Commercial Code)

(z)　In the event of a change in legal form, a merger, or a merger of companies, preparing a written report justifying these facts (Section 69b paragraph 4 of the Commercial Code)

However, this calculation should not be considered as exhaustive. There are many more duties of an executive, but we consider these to be the most important. These are obligations arising from the Commercial Code, but there are a number of related regulations that also regulate the duties of managers. According to Janakova and Zatrochova (2019), these regulations include, for example, the Commercial Register Act, the Bankruptcy and Restructuring Act, the Personal Data Protection Act/GDPR Regulation, the Competition Protection Act, the Accounting Act, and many others.

As in the case of these duties, there is no uniform catalog of the executive's rights in the case of rights either. The Commercial Code is very brief in this area and one of the basic principles of commercial law is reflected here. This principle is the principle of disposition; thus, the Commercial Code leaves it to the contracting parties to agree on all the essential rights that the executive has. In the case of an obligation, the situation is the opposite; in this case, the legislation sets a relatively large amount of the executive's obligations. This is understandable, as it ensures legal certainty and, in this form, essentially forces the manager to fulfill the requirements required by either the business company or the state authorities.

How many rights the manager will have is also affected by the commercial legal arrangement, but the main influence on the amount of rights is the contract concluded between the manager and the company. It is true that the relationship between the manager and the company in the management of the company's affairs is governed by the provisions of the mandate contract. Of course, in the event that a contract is concluded on the performance of the function of an executive or other determination of rights and obligations follows from the law, the executive will be governed by these rights and obligations established in the relevant regulations. The most general rights that every manager should have are:

The right to remuneration—in accordance with §125 par. 1 letter f) of the Commercial Code, it should be mentioned that the manager has the right to a reward, which is decided by a general meeting of the company. Of course, there may also be a situation where this reward is not agreed upon. In this case, with reference to §66 par. 6 of the Commercial Code, this will be governed by the mandate contract. In the case of remuneration for the performance of the executive function, §571 par. 1 of the Commercial Code emphasizes that, unless the amount of remuneration is agreed upon in the contract, the company is obliged to pay the manager the remuneration that is usual at the time of the conclusion of the contract for a similar activity. It is always necessary to take into account the matters that the executive arranges, because the remuneration will be different when the executive was authorized to perform only a certain range of actions (a restriction of executive authority) and when the executive has complete powers and must also arrange another agenda (Supreme Court of the Slovak Republic 2000).

The right to pay severance pay—paying severance pay is a very often discussed issue, especially in the field of labor relations. In the case of an employment contract, the Labor Code establishes the amount of severance pay that must be paid to an employee. This amount of severance pay is not regulated within the Commercial Code, but business practices include the addition of severance pay for the executive under certain conditions. The executive is entitled to severance pay only if this is explicitly stated in the contract. If it is not stated in the contract, we cannot use the Labor Code as a subsidiary (National Council of the Slovak Republic 2001).

The right to be informed—the right to information belongs to one of the basic rights that the executive has and is even established by the Commercial Code. In the performance of their function, the executive has access to a lot of information, which is often covered by the obligation of confidentiality, according to §135a par. 1 of the Commercial Code.

Information can be obtained by the executive, either through their own activity, or they can obtain information, for example, from other corporate bodies that have an obligation to inform the executive about certain facts. This information includes, for example, resolutions adopted at general meetings, the results of per rollam voting, and so on.

The right to file motions in court—in accordance with §131 of the Commercial Code, the manager has the right to file a motion in court to determine the invalidity of a resolution from a general meeting. It should be noted, however, that the number of persons listed in §131 of the Commercial Code who can file a motion to court to determine the invalidity of a resolution of a general meeting is exhaustive. If the executive believes that this resolution is in conflict with the law or another generally binding regulation, they have the right to submit this proposal. The second sentence of the first paragraph of §131 of the Commercial Code extends this right to a former executive in the event that this resolution concerns them. The illegality of the resolution itself can either lie in the contradiction of the content of the given resolution with the law, or it can lie in the method of its adoption, which would be illegal. Under the conflict of the resolution with the law, we can understand, for example, a situation where the managers would not convene the general meeting with the specified time advance. The error in the method of its adoption may consist, for example, of the fact that the shareholders decide directly at a meeting, without the consent of all the shareholders, to change the program of the general meeting.

The right to decide on the use of the reserve fund—this right arises from the provisions of §124 par. 1 of the Commercial Code, which refers to the previous provision in §67 par. 1. However, managers can use this reserve fund mainly to cover the company's losses. The reserve fund cannot be distributed among partners, nor can it be used to reward executives. The reserve fund protects the company from crisis. If the company is in crisis, this reserve fund can be used only to cover its losses.

### 4.6. Selected Duties of the Manager in the Context of Jurisprudence

As we mentioned in the previous chapters, an executive in a limited liability company has many obligations. These are obligations that are not entirely clear-cut and have been shaped over time, mainly by the decision making activity of Slovak and Czech courts.

### 4.6.1. Non-Competition

The prohibition of competition is, according to Sidak et al. (2023), a form of limiting the executive, in the sense that it limits the potential business of the executive outside the given company. This form of limitation of the duties of the manager is completely understandable, as the manager acquires a lot of information about the company while performing their duties. They obtain information about the company's contractual partners, the development of the given market, and the company's know-how. This information falls into the category of trade secrets and thus its potential disclosure to third parties or using it for your own benefit can cause damage to the company. In accordance with the Commercial Code, we can state that a manager may not conclude transactions in their own name or using their own accounts that are related to the business activity of the company.

This prohibition applies to two types of business, which are considered to be in violation of the prohibition of competition. The first type is a transaction related to the business activity of a company, which the executive has concluded in their own name. This is a situation where the executive concludes, for example, a contract in their own name, instead of concluding it in the name of the company. The second type is a transaction related to the business activity of the company, which has been concluded in the name of a third person, but on the account of the manager. In this case, an example can be understood as a contract that would not be concluded directly by the company's manager, but would be concluded by another person who would act as a commission agent for the purposes of the given transaction. The executive, in the capacity of principal, should, but not necessarily, benefit from this transaction (Skora et al. 2022).

In the context of these violations of the prohibition of competition, it should be mentioned that a conclusion of these transaction itself is a violation of the law, not the possible benefit that would flow from the contracts. Letter b of paragraph 136 par. 1 of the Commercial Code limits the executive, in the sense that they cannot mediate the company's business for other persons. That is, the manager may not perform the function of an intermediary in such deals that would normally be concluded by the company in which they perform the function of the manager.

Problematic in this case is the question of whether to also consider a situation where an executive carries out mediation free of charge as a violation of the prohibition of competition. As one of the mandatory features of the mediation contract requires remuneration, in this case, it would not be this type of contract, and, with a purely grammatical interpretation, we come to the conclusion that an executive who carries out such a mediation free of charge does not violate the law. However, such an interpretation is inadmissible from the point of view of legal doctrine. Legal doctrine, according to Patakyova and Gramblickova (2020), is inclined to the interpretation that, if certain business has been mediated by a person other than the company in which they are an executive, such a company may suffer damage, regardless of whether the executive was paid a reward or not.

Another provision of the Commercial Code is relatively clear. The executive may not participate in the business of another company as a partner with unlimited liability. In this case, it is completely irrelevant what the company's business is Matúšová and Nováček (2022). However, the unlimited liability of the manager arises only in two cases, namely, when they are to become a partner in a public company or enter into a limited partnership as a general partner. The law does not address a situation where an executive would sign a guaranty statement addressed to several creditors of the company. In this case, experts recommend always establishing a clause in the partnership agreement that would apply this non-competition clause, also to possibly guarantee, in an unlimited sense, the statements of the manager.

According to Števček and Ivančo (2021), the last limitation that applies to executives is the performance of activities as a statutory body, a member of a statutory body, or another body of another legal entity with a similar business object. Again, in this case, the company is protected, because the executive obtains a lot of valuable information during the performance of their activities, which must not be disclosed to other companies. However, the law provides an exception to this rule in the event that the company in which the executive functions is one whose business is participated in by the company. This legal provision, according to Mamojka and Ivan (2016), especially applies to parent and subsidiary companies, within which, such a potential flow of information is permitted.

The consequences of a possible violation of the prohibition of competition are governed by the general provision §65 of the Commercial Code. Provision §65 of the Commercial Code regulates the consequences of a violation of the prohibition of competition for all legal forms of commercial companies in the same way. A person (in our case, an executive) who has violated the prohibition of competition has a legal obligation to release to the company any property benefit that they acquired at the expense of the company. Property benefits can be money or things, but, at the same time, they can also be rights. Therefore, one option for the company is to request the issuance of a property benefit. The second option that the company has is to require the manager to transfer the rights corresponding to the benefit from the competitive business to the company, if the company has not yet been issued this benefit. However, both of these claims must be applied within a certain period. The subjective term is set at three months from when the company became aware of the violation of the law and the objective term is set at one year from the creation of these rights. After these deadlines have expired in vain, the company would not be able to assert these claims in court, because the deadline has a preclusive nature. In practice, this means that the claims would disappear (Matejkova and Pavelek 2019).

However, the fact that the company could exercise one of the two rights mentioned above does not mean that the company is not entitled to compensation from the executive. If

the executive commits a competitive act, then they are obliged to compensate the company for the incurred damage (actual damage + lost profit). A claim for damages, unlike the claims listed in §65 paragraph 2 of the Commercial Code, can be applied within the general four-year limitation period (Srebalová and Vojtech 2021).

Claims asserted from liability for damage caused by a violation of the prohibition of competition are governed by the amendment of lex generalis §387 et seq. Commercial Code. However, the legislation in relation to the claims listed in §65 par. 2 is considered lex specialis, and it is therefore necessary, in the case of claims under §65 par. 2 of the Commercial Code, to proceed according to these special provisions in relation to the general regulation §387 et seq. Commercial Code.

However, the duration of the executive's non-competition ban is questionable. This issue has also been dealt with by the Supreme Court of the Czech Republic and, pursuant to the Judgment dated 28 April 2010, no. 29 Cdo 20149/2009 came to the conclusion that this legal prohibition resulting from §136 of the Commercial Code for executives applies only to the period during which the given person performs the function of a statutory body. However, the court is also inclined to the interpretation that, by contractual arrangement, this prohibition can be extended to the period after the end of the position.

In conclusion, it must be said that this prohibition of competition can also be applied to partners, if so determined by the partnership agreement or articles of association. In practice, this provision is especially important for small companies, where partners are not only bound by deposits, but also ensure the normal running of the company. From the directly legal diction, it would follow that even the creation of a trade license by some of the partners is already a violation of the prohibition of competition. However, in this case, the Supreme Court of the Slovak Republic, in a Decision dated 1 January 1999, no. stamp Obo 101/99, decided that it is not a mandate of the prohibition of competition if a partner of a limited liability company obtains a trade license for an activity that is also the subject of the company's activity, as long as they do not actually perform this activity. Therefore, it is recommended that the social contract clearly defines the prohibition of competition; therefore, the term "prohibition of competition" should be understood as the very acquisition of a trade license Vítek et al. (2022).

In the case of a non-competition agreement for partners as well, one must still think about the question of whether the scope of the non-competition can be expanded or narrowed. According to the generally accepted interpretation, this scope can clearly be expanded and it would also be possible to narrow it down. Thus, partners could agree on only some letters from §136 par. 1 of the Commercial Code.

4.6.2. Proper Maintenance of Prescribed Records and Accounting

According to court the Commercial Code orders managers to ensure the proper keeping of prescribed records and accounting. The definition of this term is problematic, as the law does not contain a legal definition. However, the definition of this term is very important because, in the event that the manager does not fulfill this obligation, they may be held liable for damages (Supreme Court of the Slovak Republic 2015).

Since we do not find a clear definition of this term in the law, its definition is left to the decision making activity of the courts. In our opinion, this term is most appropriately defined by the judgment of the Supreme Court of the Czech Republic dated 10 November 1999, No 29 Cdo 1162/99, which defines when the executive is released from their liability for damage in the event of a breach of the obligation specified in §135 of the Commercial Code. In this opinion, the executive shall be relieved of their responsibility for damage caused by a violation of the obligations specified in §135 of the Commercial Code if they prove that they have ensured the maintenance of prescribed records and accounting to the necessary extent by a qualified person, for whom they have created the necessary conditions for the performance of the activity, i.e., providing the necessary cooperation (Supreme Court of the Czech Republic 1999). Based on the above, two conditions must be met cumulatively. The first condition is that the manager must select a qualified person for

the purpose of keeping the relevant records and accounting. It should be remembered that this person must be qualified not only subjectively from the executive's point of view, but also objectively from the point of view of the public and the relevant authorities. This is demonstrated in practice by taking certain courses, possibly by certain work experience, but also by belonging to a certain professional chamber. The second condition that must be met is the fact that the manager has created suitable conditions for this person to perform their activities, that is, they provided them with the materials for proper accounting. In order for the manager to be liable for damage connected with keeping records and accounting, it is sufficient that they do not fulfill at least one of these two conditions.

This interpretation is also supported by the Regional Court in Hradec Králové, which, in its Resolution dated 15 October 2001, No. 20 Co 271/2001, came to the conclusion that a defendant's breach of duty based on paragraph 135 of the Commercial Code would occur in three cases. The first case is a situation where the defendant would not ensure the proper accounting at all. The second option is a situation where the manager would ensure the accounting by another, unqualified person. Additionally, the last case would occur if the executive did not create the working conditions for the proper accounting (Regional Court in Hradec Králové 2001).

Based on the above, it follows that, in the case of a failure to maintain the proper records and accounting, three liability relationships may arise. The first responsibility relationship is the relationship between the company and the relevant authority (e.g., tax office). Subsequently, a liability relationship may arise between the company and the executive. The last responsibility relationship is the relationship between the person who managed the accounting and the professional chamber, in which, for example, disciplinary proceedings due to a breach of duties are recorded.

Defining who is a qualified person and who is not can still be problematic. The Regional Court in Hradec Králové also dealt with this issue. The Resolution dated 11 February 2003, no. 24 Co 247/2002 decided that a qualified person can be both a legal entity and a natural person who meets the qualification requirements. The qualification prerequisites consist of education and experience. From this follows the fact that education is a narrower term in terms of its content than the term qualification. Education is only one of the prerequisites for the performance of a certain activity. Qualification includes education and also experience. According to the court, administrators are responsible for choosing a person to properly manage the accounts. If the executive selects a person who is competent in this matter (has education, experience, and the method of accounting prescribed by law), then the executive has fulfilled the obligation set forth by the Commercial Code. However, in the event that the accounting is not sufficiently prepared, it is appropriate to apply the responsibility towards the person who prepared it, not towards the manager (Regional Court in Hradec Králové 2003).

### 4.6.3. Perform the Duties with Professional Care

One of the most general and, at the same time, one of the most essential obligations is the obligation of the manager to perform their duties with professional care and in accordance with the interests of the company and all its partners.

The action of the executive must therefore contain three elements, which must be fulfilled cumulatively. The first element is the executive's duty to perform their duties with professional care. The word association with professional care is mentioned in several parts of the Commercial Code, but it is not legally defined in any of these provisions. Once again, we have to use jurisprudence, which quite clearly defines this term as "the expected level of information and expertise of the manager, on the basis of which he is obliged to perform his function from the day of its creation until the day of its termination" (Judgment of the Supreme Court of the Czech Republic dated 30 November 2010, No. 39 Cdo 3376/2009) (Supreme Court of the Czech Republic). In the context of the above definition, it should be said that "knowledge" and "expertise" are not associated with the subjective abilities of the executive, but are objective personal qualities that every person

acting as an executive should meet. That is, if the executive acts with the highest possible professional care of which they are subjectively capable, but from an objective point of view it would be insufficient, then this is an action contrary to the provisions of §135a par. 1 of the Commercial Code. The second element that must be fulfilled is acting in accordance with the interests of the company. The last, third element is acting in accordance with the interests of all the partners. Based on the legal diction "in accordance with the interests of the company and all its partners", it may appear that the interests of the company are equivalent to the interests of all the partners. Legal theory, however, has an opposite opinion on this issue, in the sense that the interests of the company should come first and the interests of the shareholders second. This is based on the fact that the interests of partners are often different from the interests of the company. As an example, it is possible to cite decisions on the division or non-distribution of profits. It would be in the interest of the company not to divide the profit among the partners, but to use this profit for the further development of the company. On the other hand, the partners will try to ensure that the profit is just divided between them and not invested back into the company. Therefore, the manager's decision making on this matter is not easy at all, and the manager must carefully consider what proposal for the distribution of profit they will present to the general meeting. Several authors are inclined to the interpretation that the executive should act so that every decision is in line with the company's interests first and foremost, even in a situation where it conflicts with the individual interest of one of the shareholders (Mamojka and Ivan 2016).

These three elements of the procedure are subsequently referred to in the second sentence of §135a par. 1 of the Commercial Code. This is a demonstrative calculation expressed by the word "especially" and there are therefore more obligations that are linked to the mentioned three elements of this procedure. The law specifies the following obligations:

1. To obtain and take into account all the available information regarding the subject of the decision when making a decision,
2. To maintain confidentiality about confidential information and facts, the disclosure of which to third parties could cause damage to the company or threaten its interests or the interests of its partners,
3. When performing their duties, managers must not prioritize their interests, the interests of only some partners, or the interests of third parties over the interests of the company.

Add. (1) Once again, even in this case, an objective effort of the manager is required to obtain and take into account all the available information when making a decision. Commonly available information is considered to be information available from various registers (Business Register of the Slovak Republic) and from courts about ongoing proceedings in which there is a partner with whom the manager wants to enter into a contract with.

Add. (2) Maintaining confidentiality is very important, because the executive obtains a lot of information during the performance of their activities, which could potentially harm the company. Although the given provision of the law does not contain the obligation of the executive to maintain confidentiality about trade secrets, this obligation is imposed on everyone who has the right to dispose of such a secret. The executive must observe confidentiality about confidential information and, at the same time, about trade secrets. What is considered to be confidential information is defined in §271 of the Commercial Code. Confidential information is considered to be information provided to each other by the parties negotiating the conclusion of a contract that is marked as confidential. Continued text in §135a par. 1, however, expands upon the range of protected information. The executive must observe confidentiality regarding trade secrets and confidential information, as well as facts that could cause damage to the company if disclosed to third parties. The last sentence of the aforementioned provision is the most general, and therefore the manager must consider which information they could potentially disclose and which information they must not disclose.

Add. (3) The executive must perform their duties in such a way that they prioritize the interests of the company. They must not prioritize their own interests and, as stated in the previous parts of the text, they must not prioritize the interests of the partners either, if they conflict with the interests of the company.

## 5. Discussion and Conclusions

In our scientific study, we provided, in our opinion, a fairly comprehensive view of the manager as a statutory body of a limited liability company. It is important to deal with this issue, especially because limited liability companies are the most common form of businesses, not only in the territory of the Slovak Republic, but also in other countries. There can be no doubt that this feature is clearly one of the most important features. We base this claim on the fact that, in contrast to the supervisory board, it is a mandatory position that every limited liability company must have filled, and we can therefore say with certainty that the position of manager is irreplaceable.

In the introductory part of this study, due to a multidisciplinary investigation of this issue, we paid attention to the definition of managerial knowledge and skills from the point of view of managerial theory. However, while the manager should have these managerial knowledge and skills, but they do not have to have them. In the following chapter, however, we examined, in detail, the legal conditions that a natural person must meet if they want to perform the function of an executive (Funta and Králiková 2022).

This scientific study provides an overview of the position of the manager and their scope and relatively comprehensively solves the issue of the business obligations between a manager and their limited liability company. The entire work is supplemented by substantial jurisprudence from the judiciary of the Slovak and Czech Republics. Using possible methods, we tried to provide the most comprehensive overview of the rights and obligations of executive function.

As part of the investigation of the relationship between the manager and the limited liability company, with a focus on the management of the business company, we did not encounter any serious contradictions. In general, this relationship is almost exclusively subject to the regime of the Commercial Code, most often on the basis of a mandate contract. Some issues can be problematic, such as the payment of severance pay upon a termination of the position of an executive, where it is not possible to apply the provisions of the Labor Code because the executive is not an employee of a limited liability company but, based on a mandate agreement, its contractual partner. The legislation allows the manager of a limited liability company to be an employee of this capital trading company, but not in the position of a manager and only as an ordinary employee.

However, we positively evaluated the impact of jurisprudence, which appropriately fills the legislative gaps in the position of the executive in relation to the limited liability company, thus contributing to the legal certainty of the contracting parties.

As part of the answer to the research question, we are of the opinion that the performance of the function of the executive, especially their rights and obligations, is sufficiently regulated in the Slovak legal order despite the fragmentation, and there is no need for legislative intervention by legislators.

The following key findings emerged from our analysis of a manager's rights and obligations. The manager in a limited liability company is one of the most important functions and they have a wide range of rights and responsibilities. Such a wide range of rights and duties also requires a high level of responsibility. Managers are one of the best paid employees of individual companies and they are therefore also required to be highly professional and competent, which is reflected in the Commercial Code in the form of this wide range of rights and obligations. On the other hand, their responsibility is also at a much higher level compared to that of ordinary employees. The key findings of this article, in relation to the above, are, in particular, that the manager acts as a legal representative of the limited liability company and their actions are considered to be a manifestation of the company's will. This emphasizes the importance of understanding the manager's

role and the legal implications of their actions. The rights and obligations of the manager are determined by the statutory provisions in the Commercial Code and the Civil Code. These provisions outline the scope of the manager's authority, their responsibilities, and the limits on their actions. Law requires the manager to be a natural person, and it is generally more appropriate if the manager is also a partner of the limited liability company due to their psychological commitment to the company's benefit. However, the manager must also possess the necessary managerial skills to effectively carry out their responsibilities. These insights highlight the significance of the legislative basis in defining the rights and obligations of the manager in a limited liability company.

The analysis of the rights and obligations of the manager of a limited liability company revealed that the Slovak legal order provides a comprehensive framework for governing these managerial activities. The Commercial Code and the Civil Code outline the key provisions governing the role of the manager, including their rights, obligations, and the limits on their actions. These legal provisions offer a clear legislative basis for the functioning of managers within the limited liability company structure. The only exception is the absence of qualification (educational) prerequisites for the performance of the executive function.

The strengths of these existing regulations include the clarity, specificity, and flexibility provided by the articles of association for the protection of company interests. The legal provisions provide specific guidelines and requirements for the appointment, rights, and obligations of managers. This clarity helps to ensure that managers understand their roles and responsibilities, reducing the potential for confusion or disputes. The ability to modify the actions and limitations of individual managers through the articles of association provides flexibility for companies to tailor the managerial framework to their specific needs and circumstances. This allows for the customization of and adaptation to the unique characteristics of each company. The legal provisions set out safeguards to protect the interests of limited liability companies. For example, the requirement that managers be natural persons and not listed as obligees in the register of authorizations for execution helps to ensure that only individuals with the necessary qualifications and integrity can assume managerial roles.

However, it is important to acknowledge the weaknesses or potential areas for improvement in these existing regulations, such as the limited guidance on managerial qualifications, potential gaps in oversight and control, and the practical challenges of implementation. While the law mandates that managers possess the necessary qualifications, there is a lack of explicit criteria or guidelines for assessing these managerial skills. This may lead to inconsistencies in the evaluation and appointment of managers, potentially impacting the effectiveness of their performance. The current regulations do not provide detailed mechanisms for monitoring or regulating managerial activities. While the roles of partners or stakeholders in controlling the actions of managers are acknowledged, there may be a need for more robust mechanisms to ensure proper accountability and prevent abuses of power. Despite the comprehensive legal framework, practical challenges may arise in the application and enforcement of these regulations. These could include difficulties in assessing the suitability of managers, ensuring compliance with legal requirements, or resolving disputes related to managerial decisions.

In conclusion, while the existing legal framework in the Slovak legal order provides a solid basis for regulating the rights and obligations of the manager in a limited liability company, there are areas where improvements could be made. Addressing the potential weaknesses by offering clearer guidelines on managerial qualifications, enhancing oversight mechanisms, and addressing practical implementation challenges would further strengthen the effectiveness of these regulations. Overall, the existing legislative basis provides a foundation for the proper functioning of managers in limited liability companies, but ongoing evaluations and potential legislative interventions should be considered to ensure the continued relevance and effectiveness of these regulations.

In this context, however, we consider it necessary to state that the limitation of our research was the legislative state, which, unfortunately, does not regulate the qualification

conditions for the performance of the function not only of the executive of a limited liability company, but also for the statutory body of a commercial company.

As part of our investigation, it is recommended that legislators also focus their attention on the question of the qualifications of the statutory body of a commercial company, which has not been addressed and investigated by anyone so far. This issue will be the subject of our further research. We assume that we will be able to propose an amendment to the Commercial Code with such provisions that would, in a mandatory manner, urge qualification requirements for the statutory bodies of commercial companies, as is the case in public law sectors.

**Author Contributions:** Conceptualization, M.K. and T.P.; methodology, M.K.; software, T.P.; validation, M.K. and T.P.; formal analysis, M.K. and T.P.; investigation, M.K.; resources, T.P.; data curation, M.K.; writing—original draft preparation, M.K. and T.P.; writing—review and editing, T.P.; visualization, T.P.; supervision, T.P.; project administration, T.P.; funding acquisition, T.P. All authors have read and agreed to the published version of the manuscript.

**Funding:** This research was funded by project National infrastructure for supporting technology transfer in Slovakia II–NITT SK II, co-financed by the European Regional Development Fund.

**Institutional Review Board Statement:** Not applicable.

**Informed Consent Statement:** Not applicable.

**Data Availability Statement:** The data used, especially legal regulations, can be found on the website of the Collection of Laws of the Slovak Republic: www.slov-lex.sk.

**Conflicts of Interest:** The authors declare no conflict of interest.

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
