# Peer review of "A Critical Analysis of the Rights and Obligations of the Manager of a Limited Liability Company: Managerial Legislative Basis"

_laws, 2023_

Round 1

Reviewer 1 Report

The author in the paper, in addition to the analysis of the manager as a statutory body of a limited liability company, focus on a critical analysis of the rights and obligations of the manager.

The article critically evaluate the results of the investigation, compared the development of Slovak and Czech jurisprudence and its impact on the subject under investigation. The paper was able

to give answer the research question whether the position of the executive as a statutory body of a limited liability company is sufficiently regulated or if a significant legislative intervention in the Slovak legal system is needed.

The paper is well structured, the methods used in the article, such as analysis, synthesis, comparison, deduction, description, were well performed. The summary of literature is adequate to the research.

I give credit to the author for a unique take on the critical analysis of the rights and obligations of the manager of a limited liability company.

Author Response

Dear Reviewer 1,

thank you very much for your positive review, which we greatly appreciate.

Best regards, authors

Reviewer 2 Report

Abstract
The abstract lacks specific details about the research findings, implications of the analysis, and the conclusive answer to the research question, which makes it difficult to assess the significance and impact of the study entirely.
Introduction
The introduction lacks a clear statement of the specific research problem or objectives the study aims to address. While the introduction provides general background information on limited liability companies, the role of managers, and the need for effective legal regulation, it does not clearly define the research gap or the specific focus of the study. Without an apparent research problem or objective, it becomes difficult for readers to understand the purpose and significance of the research.
Literature Review
It does not establish a clear connection between the various viewpoints or present an organized narrative that leads to the research problem or objective. Providing a more structured review highlighting the key themes, debates, or gaps in the existing literature and how they relate to the research topic would be beneficial.
Aim, Materials, and Methods
  This section lacks detail and clarity in describing the specific materials and methods employed in the study. While the general approach is mentioned, such as the analysis of managerial knowledge and skills, individual legal provisions, and the comparison of Slovak and Czech jurisprudence, the specific procedures and techniques used are not clearly outlined. It would be beneficial to provide more detailed information on how the analysis and comparison were conducted, such as the selection criteria for legal provisions or the method of identifying and analyzing relevant court judgments.
  Additionally, the section could benefit from elaborating on the rationale behind choosing the selected legislative acts and databases as the main sources of knowledge. Providing a brief explanation of why these sources were considered relevant and reliable for the study would enhance the clarity and credibility of the research.
  Furthermore, while the use of logic, abstraction, synthesis, analytical method, and comparison method is mentioned, their application to the study is not explained in detail.
Result part: I don’t understand it totally
Discussion and Conclusion
  The limitation of this part is that it lacks a specific and detailed discussion of the findings and results of the study. While it is mentioned that a comprehensive view of the manager as a statutory body of a limited liability company was provided, there is no specific mention of the key insights or conclusions drawn from the analysis of managerial rights and obligations.
  Furthermore, while the research question is addressed, stating that the performance of the executive's function is sufficiently regulated in the Slovak legal order, no detailed explanation or argumentation is provided to support this conclusion. It would be valuable to include a more thorough discussion of the reasons behind this assessment, considering the strengths and weaknesses of the existing regulations and their practical implications.

Minor editing of the English language required

Author Response

Dear Reviewer 2,

Thank you very much for your constructive opinion, which we greatly appreciate. Using the document change tracking system, it is possible to see, but especially to assess, that we respected and incorporated all your comments, which also caused an increase in the number of manuscript pages.

Best regards, authors

Reviewer 3 Report

.

Author Response

Dear Reviewer 3,

Thank you very much for your constructive opinion, which we greatly appreciate. Using the document change tracking system, it is possible to see, but especially to assess, that we respected and incorporated your comments, which also caused an increase in the number of manuscript pages.

We paid the main attention to the expansion of the introduction, the methodology as well as the discussion with the conclusion, where we specified the main contribution of the article.

The revised version also includes reviewer 2's comments.

We believe that the manuscript edited in this way meets your expectations of a high-quality scientific article that also has a benefit.

Best regards, authors

Round 2

Reviewer 2 Report

Section 3 should be Materials and Method used

Discussion and conclusion should be supported by previously conducted. At the same time, limitations and future research direction should be included

The study recommendations should be added.

 Minor editing of English language required

Author Response

Dear Reviewer 2,

thank you for your comments, which we respected. We used the change tracking system again and marked the changes made based on the reviewers' recommendations in green:
- we have edited Chapter 3
- we have added the recommendations as well as the limitations of our scientific study as well as the intended subject of future research, 

- we checked the english language

Sincerely,  Authors

Reviewer 3 Report

The authors take into consideration all the suggestions made in the first phase.

Author Response

Dear Reviewer 3,

thank you for your comments, which we respected. Again, we used the change tracking system and marked the changes made in green based on the recommendation of all reviewers, namely:
- we edited Chapter 3
- we have added the recommendations as well as the limitations of our scientific study as well as the intended subject of future research.
- we checked the english language,
- we have added references

Sincerely, Authors